# Fine-tuning with RAG for Improving LLM Learning of New Skills

## Abstract

Large language model (LLM) agents deployed for multi-step tasks frequently fail in predictable ways: attempting actions with unmet preconditions, issuing redundant commands, or mishandling environment constraints. While retrieval-augmented generation (RAG) can improve performance by providing runtime guidance, it requires maintaining external knowledge databases and adds computational overhead at every deployment. We propose a simple pipeline that converts inference-time retrieval into learned competence through distillation. Our approach: (1) extracts compact, reusable hints from agent failures, (2) uses these hints to generate improved teacher trajectories via one-shot retrieval at episode start, and (3) trains student models on these trajectories with hint strings removed, forcing internalization rather than memorization. Across two interactive benchmarks, ALFWorld (household tasks) and WebShop (online shopping), distilled students consistently outperform baseline agents, achieving up to 91% success on ALFWorld (vs. 79% for baselines) and improving WebShop scores to 72 (vs. 61 for baselines), while using 10-60% fewer tokens than retrieval-augmented teachers depending on the environment. The approach generalizes across model scales (7B/14B parameters) and agent architectures (ReAct/StateAct), demonstrating that retrieval benefits can be effectively internalized through targeted fine-tuning without permanent runtime dependencies.

## 1 Introduction

Large language models are increasingly deployed as agents that interact with environments to complete multi-step tasks. Success requires not just generating plausible text but maintaining goals across extended interactions, managing state and preconditions, and recovering from errors. Despite advances in prompting strategies, LLM agents still exhibit systematic failures in interactive environments.

Prior work has explored multiple approaches to improve agent performance. Structured prompting methods like ReAct (Yao et al., 2023b) and StateAct (Rozanov & Rei, 2025) provide scaffolding for reasoning and state tracking. Self-reflection approaches such as Reflexion (Shinn et al., 2023) enable learning from mistakes across multiple attempts. Retrieval-augmented methods (Lewis et al., 2021; Zhao et al., 2024; Fu et al., 2024) inject external knowledge to guide decisions. Fine-tuning approaches (Chen et al., 2023; Zhang et al., 2025) directly modify model parameters using expert demonstrations. Each approach involves trade-offs: retrieval adds token cost and deployment complexity, fine-tuning risks overfitting and requires substantial data, while self-reflection assumes multiple attempts are feasible.

We propose a pipeline that combines the benefits of retrieval and fine-tuning while avoiding their individual limitations. Our key insight is that retrieval-augmented generation need not remain a permanent runtime dependency. Instead, it can serve as a source of improved training supervision that gets internalized into model parameters. Specifically, we: (1) run base agents to collect failures, (2) extract generalizable hints from these failures, (3) use hints to generate better teacher trajectories, and (4) distill these trajectories into students that no longer need hints.

Our contributions are:

- Failure-driven hint extraction: A method to automatically generate typed, reusable guidance from agent mistakes without expert supervision, using GPT-4o to diagnose failures and propose corrective rules

- One-shot retrieval distillation: A training approach where hints are retrieved once at episode start during teacher data generation, then removed during student training to force internalization

- Efficiency analysis: Demonstration that distilled models dominate the accuracy-efficiency frontier, achieving highest task success with lowest token usage

## 2 RELATED WORK

### 2.1 PROMPTING-BASED AGENTS

Chain-of-thought prompting (Wei et al., 2022) revealed that explicit reasoning steps improve complex task performance. ReAct (Yao et al., 2023b) extended this to interactive settings by interleaving "Thought" and "Action" tokens, achieving strong results on ALFWorld and WebShop without training. StateAct (Rozanov & Rei, 2025) augmented ReAct with explicit state representations (goal, inventory, location) that ground reasoning over long horizons, improving success rates by 10-30% across environments. THREAD (Schroeder et al., 2025) introduced recursive reasoning through dynamically spawned sub-threads, allowing agents to allocate computation to complex subtasks. These methods demonstrate impressive zero-shot capabilities but remain limited by the parametric knowledge of the model.

### 2.2 LEARNING FROM EXPERIENCE

Several approaches enable agents to learn without gradient updates. Reflexion (Shinn et al., 2023) generates self-critiques after failures and accumulates lessons across multiple attempts on the same tasks, reaching 91% on HumanEval through iterative refinement. However, this requires ground-truth feedback and multiple trials per task. ExpeL (Zhao et al., 2024) extracts insights from training trajectories and retrieves them alongside demonstrations at test time, improving systematically as experience grows. AutoGuide (Fu et al., 2024) learns state-conditioned guidelines that are retrieved dynamically based on current context. AutoManual (Chen et al., 2024) builds comprehensive domain manuals through exploration. While effective, these methods create permanent dependencies on external knowledge stores and retrieval mechanisms.

### 2.3 FINE-TUNING AND DISTILLATION

FireAct (Chen et al., 2023) demonstrated that fine-tuning on just 500 GPT-4 trajectories can improve a 7B model's success rate by 77% on tool-use tasks. Parameter-efficient fine-tuning has been widely studied (Hu et al., 2021; Dettmers et al., 2023; Han et al., 2024), enabling adaptation of large backbones with small memory and compute cost. Prompt distillation (Kujanpää et al., 2025) showed that complex prompts can be compressed into model weights by training on outputs generated with the prompt but without providing it as input. Chain-of-thought distillation (Hsieh et al., 2023) and self-training approaches (Zelikman et al., 2022) transfer reasoning traces into smaller models. Classical model distillation approaches such as DistilBERT (Sanh et al., 2020) compress large transformers into lighter students, but operate in a static supervised setting rather than interactive environments. Our work differs from these approaches: unlike FireAct which requires expensive GPT-4 supervision, we generate teacher data from the model's own failures augmented with self-extracted hints. We distill dynamically retrieved, failure-specific guidance rather than standard prompt distillation which compresses static prompts.

### 2.4 POSITIONING OUR APPROACH

Unlike multi-attempt approaches, we learn from a single training run, and we use retrieval only during training, not as a permanent fixture attached to the model. We also evaluate using a single

run on the test set. We automatically generate supervision from the failures of the agent rather than requiring expert demonstrations. This positions our method as a practical path to improving deployed agents without runtime complexity.

# 3 METHODOLOGY

## 3.1 PROBLEM SETTING

We model tasks as partially observable MDPs $(\mathcal{S}, \mathcal{A}, T, R, \Omega, O)$ with latent state $s_t \in \mathcal{S}$, observation $o_t \in \Omega$ drawn from $O(\cdot \mid s_t)$, action $a_t \in \mathcal{A}$, transition $T(s_{t+1} \mid s_t, a_t)$, and reward $r_t = R(s_t, a_t)$.

We define different agent policies:

- **Base policy**: $\pi_\theta(a_t \mid o_{\leq t})$ conditions only on the observation history.
- **RAG policy**: $\pi_\theta^{\text{RAG}}(a_t \mid o_{\leq t}, H_0)$ where $H_0 = \{h_1, \ldots, h_k\}$ are hints retrieved once at $t = 0$.
- **SFT policy**: $\pi_\phi^{\text{SFT}}(a_t \mid o_{\leq t})$ with parameters $\phi$ fine-tuned on successful trajectories from $\pi_\theta$ (without retrieval).
- **Distilled policy**: $\pi_\phi(a_t \mid o_{\leq t})$ with parameters $\phi$ trained on successful trajectories from $\pi_\theta^{\text{RAG}}$, with hint strings removed to enforce internalization.

For ReAct agents, we augment the action space to include reasoning: $a_t = (c_t^{\text{thought}}, c_t^{\text{action}})$ where $c_t^{\text{thought}}$ is internal reasoning and $c_t^{\text{action}} \in \mathcal{A}$ is the environment action. StateAct further augments with explicit state: $a_t = (c_t^{\text{state}}, c_t^{\text{thought}}, c_t^{\text{action}})$. We evaluate on two complementary benchmarks:

ALFWorld (Shridhar et al., 2021): A text-based household environment with 6 task types (Pick & Place, Examine in Light, Clean & Place, Heat & Place, Cool & Place, Pick Two & Place) requiring multi-step object manipulation. Agents navigate rooms, operate containers, and transform objects using commands like `goto`, `take`, `open`, `clean`.

WebShop (Yao et al., 2023a): An online shopping environment with 1.18M real products where agents must find and purchase items matching natural language specifications. Agents navigate search results, compare products, and select options using actions like `search[query]`, `click[element]`. For WebShop specifically, we omit the $c_t^{\text{thought}}$ component in ReAct & State-Act, consistent with prior findings that this leads to stronger prompt-only performance in that environment (Rozanov & Rei, 2025).

## 3.2 PIPELINE OVERVIEW

Our training pipeline (Figure 1) consists of four stages:

**Stage A - Base Agent Rollouts:** We deploy base agents (ReAct or StateAct) across the training split, collecting both successful and failed trajectories. Successes form the baseline supervised fine-tuning (SFT) dataset; failures serve as input for hint extraction.

**Stage B - Self-Hint Extraction:** For each failed trajectory $\tau$, we construct a complete failure example containing the task instruction, initial observation, full action sequence, and resulting observations. We then prompt GPT-4o to generate 1-4 imperative hints that diagnose the failure and provide generalizable guidance. Hints use placeholders ({object}, {container}) and are typed by task category for retrieval. Some examples of extracted hints:

- Ensure the {container} is open before attempting to place the {object} inside.
- Verify inventory capacity before attempting to take additional items.
- Use a systematic search pattern to avoid missing {object} in {location}

**Stage C - Teacher Data Generation:** Given instruction $g$ and initial observation $o_0$, we determine the task category $c(g)$ (explicitly provided in ALFWorld or inferred from the instruction in WebShop) and retrieve: $H_0 = \arg\text{top-k}_{h \in \mathcal{H}_{c(g)}} s_\psi(h; g, o_0)$,

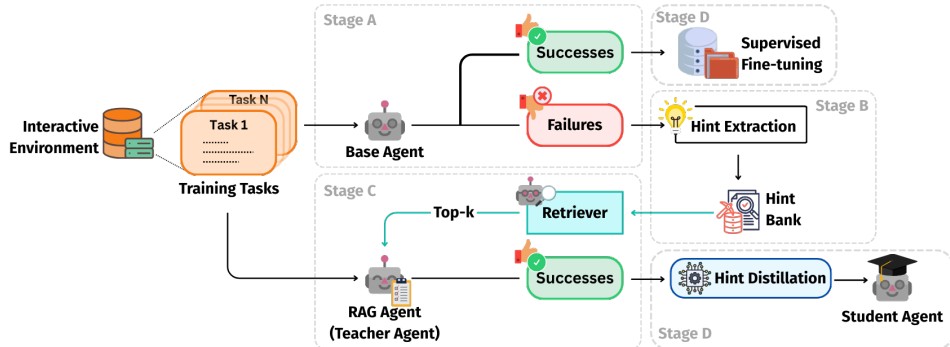

Figure 1: Training Pipeline. Stage A represents the initial base agent run. Stage B is the hint extraction using failures of the agent. Stage C is running the RAG agent. Stage D is training the models based on regular supervised fine-tuning (SFT) and our distillation method.

where $s_\psi$ is an LLM re-ranking score computed by a quantized Qwen-2.5 7B (Qwen et al., 2025) model that evaluates candidate hints given the task instruction $g$ and initial observation $o_0$. We inject $H_0$ once at $t=0$ and then run the teacher $\pi_\theta^{\text{RAG}}(a_t \mid o_{\leq t}, H_0)$ for the remainder of the episode.

The addition of this one-time hint block equips the agent with corrective priors distilled from earlier failures (from the *training set*). By running the teacher policy in this retrieval-augmented mode, we collect a new set of successful trajectories that represent improved behaviors relative to the baseline. Importantly, only *successful* episodes are retained at this stage, since the purpose of the teacher is to provide high-quality demonstrations for distillation.

**Stage D - Dataset Construction and Training:** Datasets are constructed for supervised fine-tuning (SFT) from the collected trajectories. Two complementary sets are defined:

1. **Base Dataset (SFT)**: $\mathcal{D}_{\text{SFT}}$ composed of successful agent training trajectories produced by the base agent from Stage A, representing the baseline policy without retrieval.

2. **RAG Dataset (Distillation)**: $\mathcal{D}_{\text{Distillation}}$ composed of successful agent+RAG trajectories produced by the teacher from Stage C, with the hint strings removed from the serialized text. This forces the student model to internalize the hints implicitly, rather than memorizing them.

We then train low-rank adapters (LoRA) (Hu et al., 2021) on teacher trajectories with hint strings removed from inputs. This forces students to learn behaviors that substitute for explicit hints rather than memorizing the text.

Together, these four stages form a closed feedback loop: failures of the base agent are mined for corrective hints, which in turn guide the teacher policy, yielding stronger demonstrations for student training. This pipeline operationalizes the idea of self-improvement without reliance on external experts or handcrafted rules.

### 3.3 HINT EXTRACTION DETAILS

To reduce reliance on hand-engineered rules, we induce compact, reusable hints directly from the failures of our own base agents. Our extraction process maps failed rollouts to compact hints, which are partitioned by task type or product category. The goal of this process is to distill actionable advice that can be reused across similar tasks, thereby improving robustness without requiring privileged supervision. We design prompts for GPT-4o that enforce constraints on hint length, form, and structure: hints must be imperative, use placeholders for generality, and emphasize preconditions and sequencing rather than surface descriptions. Outputs are required to be strict JSON for programmatic parsing.

Extracted hints are then normalized, deduplicated using fuzzy matching, and grouped by task type (ALFWorld) or product category (WebShop), enabling efficient retrieval within relevant partitions. Deduplication uses fuzzy string matching with a Levenshtein distance threshold of 0.85, chosen

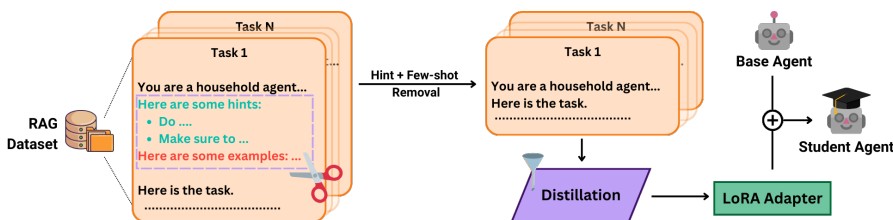

Figure 2: Hint distillation process. We remove the hint block and few-shot examples, since these are constant per task and provide no useful training signal. The end result is a trained adapter. During inference, we combine the adapter with the base agent to get our trained student.

empirically to catch near-duplicates (e.g., word variations) while preserving semantically distinct hints. The prompt is shown in Appendix A.1.

### 3.4 RETRIEVAL MECHANISM

At $t=0$, we retrieve a fixed set of $k$ hints $H_0$ from the category-specific bank, conditioned on the task instruction $g$ and initial observation $o_0$. No additional retrieval occurs during the episode, the agent must rely on this static set of hints for the remainder of the trajectory. This design both limits token overhead and reflects a realistic use case where guidance is provided only once at the outset rather than continuously. The retriever: (i) determines the task category from the instruction (explicitly provided in ALFWorld, inferred in WebShop), (ii) scores candidate hints within the relevant partition using an LLM re-ranking step (Sun et al., 2023; Qin et al., 2024) (with a quantized Qwen-2.5 7B model), and (iii) selects the top-$k$ hints to form the hint block. The block is injected immediately after the task description and before the few-shot examples. The full agent prompt is provided in Appendix A.3.

### 3.5 TRAINING CONFIGURATION

We construct two datasets from successful agent rollouts: (i) $\mathcal{D}_{\text{SFT}}$ from the baseline agent and (ii) $\mathcal{D}_{\text{Distillation}}$ from the retrieval-augmented teacher with all hint strings and few-shot demonstrations removed from the trajectory $\tau$. Since few-shot demonstrations are fixed across tasks and do not reflect the agent's actual decision process, we exclude them to prevent the student from memorizing constant scaffolds rather than learning the underlying behaviors.

We train our student on $\mathcal{D}_{\text{Distillation}}$, and as an ablation, we also train it on $\mathcal{D}_{\text{SFT}}$ to compare between baseline supervised fine-tuning with our distillation method. We fine-tune low-rank adapters (LoRA) inserted into attention and MLP projections of the backbone while freezing all base weights. Let $x_{1:N}$ denote the tokens of a serialized teacher trajectory $\tau$. The training objective is full-sequence next-token cross-entropy:

$$\min_{\phi} \ \mathbb{E}_{x \sim \mathcal{D}} \left[ -\frac{1}{N} \sum_{i=1}^{N} \log \pi_{\phi}(x_i \mid x_{<i}) \right], \tag{1}$$

where $\mathcal{D} \in \{\mathcal{D}_{\text{Distillation}}, \mathcal{D}_{\text{SFT}}\}$ and $\pi_{\phi}$ is the student with parameters $\phi$. This is equivalent to maximum likelihood training on teacher demonstrations and can be interpreted as distillation from a retrieval-augmented teacher.

As we perform fine-tuning with LoRA on top of frozen backbones, we only optimize adapter parameters attached to attention and MLP projections while keeping base weights fixed. This preserves general language competence and yields stable updates with a small memory footprint.

Each episode is serialized as a single sequence (system + instruction + initial observation followed by step turns). For Distillation, we remove the hint block from the input to prevent copying. This allows us to internalize the hints into the model weights, ultimately letting the agent reproduce teacher behavior without hints at inference (as seen in Figure 2). Training minimizes the token-level cross-entropy in Eq. 1. We apply token-level label smoothing for WebShop to mitigate overconfidence on its shorter traces. ALFWorld has a step limit of 50, so its traces are much longer than WebShop's 15 step limit.

We obtain a student policy $\pi_\phi$, where $\phi$ denotes the backbone parameters $\theta$ (that remain frozen throughout fine-tuning) together with the low-rank adapter $\Delta(\phi)$ applied on top. In other words, $\phi$ is $\theta$ with LoRA adapters integrated. $\pi_\phi(a_t|o_{\leq t}) \approx \pi_\theta^{\text{RAG}}(a_t|o_{\leq t}, H_0)$

# 4 EXPERIMENTAL SETUP

## 4.1 DATASETS AND EVALUATION

We use the standard `train`/`test` partitions and fixed step/context budgets summarized in Table 1.

Table 1: Datasets and evaluation budgets. **Note:** Success in WebShop does not account for partial fulfillment, only scores of 100 are considered successful.

| Env | #Episodes | Max Steps | Obs. | Actions | Success |
|---|---|---|---|---|---|
| ALFWorld | 1200 Train/134 Test | *50* | text | verb–object | goal reached |
| WebShop | 1200 Train/100 Test | *15* | text/HTML | search/click | score = 100 |

We use the standard splits provided by the environment. ALFWorld provides a train set of size 3553, and an unseen test set of 134 samples. We use the first 1200 samples from the train set in our experiments, and the full 134 unseen test set for evaluation. The unseen set consists of new room layouts and object-item combinations different from those in the train set.

For WebShop, we take the first 1200 examples out of the original train set of 10,587 as our training set. We do the same with the test set, we take the first 100 examples out of 500 total. The reason behind picking the first 100 was to cap the compute costs. We provide episode IDs in the trials to assist in reproducibility, all subsets are seeded and reproducible. The details of the splits and distribution of task categories are shown in Table 2.

To isolate the effect of retrieval, we construct two training sources. (i) SFT uses successful trajectories from the base agent (without hints), yielding a hint-free policy. (ii) Distillation uses successful trajectories produced by the retrieval-augmented teacher (Agent+RAG), but we strip the hint strings from the input serialization so the student learns the induced behavior rather than relying on hints. For WebShop we retain high-scoring episodes (score $\geq 67$) to ensure a sufficient, clean training set.

Table 2: ALFWorld & WebShop data split distribution and categories

| ALFWorld | | | WebShop | | |
|---|---|---|---|---|---|
| Task Category | Train set | Test set | Product Category | Train set | Test set |
| Cool & Place | 159 | 21 | Beauty | 262 | 24 |
| Clean & Place | 248 | 31 | Electronics | 219 | 19 |
| Examine in Light | 104 | 18 | Fashion | 251 | 23 |
| Heat & Place | 152 | 23 | Food | 239 | 20 |
| Pick & Place | 258 | 24 | Furniture | 229 | 14 |
| PickTwo & Place | 279 | 17 | | | |
| **Total** | 1200 | 134 | **Total** | 1200 | 100 |

## 4.2 BASELINES AND VARIANTS

We evaluate four policy families under matched decoding and step budgets, holding prompt scaffolds fixed within each agent family so that differences reflect retrieval and distillation rather than prompt engineering.

1. **Prompt-only baselines (Base).** ReAct interleaves Thought and Action with no external guidance. StateAct augments the prompt with Goal/State notes. We use both to assess

whether our process transfers across prompting styles (for WebShop, we omit "Thought" tokens for both agents).

2. **Retrieval at inference (RAG).** ReAct+RAG and StateAct+RAG receive a fixed advisory block of the top-$k=3$ hints injected once at $t=0$. Hints are advisory and must yield to the current observation in case of conflict.

3. **Supervised fine-tuning (SFT).** Trained on successful prompt-only Agent training set trajectories without hints and evaluated without retrieval.

4. **Distillation.** Trained on successful Agent+RAG trajectories with hint strings removed, and evaluated without retrieval to test whether the student internalizes the guidance.

All policies are run with **Qwen-2.5 14B Instruct** and **Qwen-2.5 7B Instruct** backbones under identical decoding settings, step caps, and seeds across comparisons.

### 4.3 IMPLEMENTATION DETAILS

For hint induction, we used GPT-4o to analyze all failed trajectories, yielding 760 ReAct and 650 StateAct hints in ALFWorld, and 756 ReAct and 831 StateAct hints in WebShop after deduplication.

When calculating token cost, we also include the quantized 7B retriever and the injected hint block (100 tokens for k=3 on average).

To ensure data quality, we filtered out trajectories with more than two invalid actions or repeated no-ops, preventing the student from imitating incorrect behaviors.

### 4.4 TRAINING PROTOCOLS

We train student models using a QLoRA-style (Dettmers et al., 2023) setup: the backbone is quantized to 4-bit, while LoRA adapters are trained in 16-bit precision (bf16). We use Unsloth (Han et al., 2023) for adapter loading and low-precision execution, and TRL's `SFTTrainer` (von Werra et al., 2020) for supervised fine-tuning. Environment-specific hyperparameters appear in Table 3.

Table 3: Training hyperparameters. LR is the learning rate, Seq. len is the maximum input sequence length during training, LoRA rank is the dimension of the low-rank adapters, and LoRA $\alpha$ is the scaling factor applied to them.

| Environment | LR | Seq. len | LoRA rank | LoRA $\alpha$ | Dropout | Weight decay |
|---|---|---|---|---|---|---|
| ALFWorld | 2e-4 | 1024 | 64 | 128 | 0.10 | 0.01 |
| WebShop | 2e-4 | 1024 | 16 | 32 | 0.20 | 0.05 |

For optimization, we use 8-bit AdamW (Loshchilov & Hutter, 2019) with a linear schedule, batch size 2 and 4 gradient-accumulation steps (effective batch size 8), a single training epoch, and 10% warm-up. WebShop uses token-level label smoothing ($\varepsilon=0.1$) to reduce overconfidence on short traces, ALFWorld uses $\varepsilon=0$. All experiments run on a single NVIDIA A100 (80 GB) with a seed of 42.

## 5 RESULTS

### 5.1 MAIN PERFORMANCE RESULTS

Table 4 shows performance across methods and model scales. Distilled students achieve highest success without retrieval, validating our core hypothesis.

In ALFWorld, distillation exceeds RAG at 14B (91.04% vs. 82.09%) and improves over both base and SFT; at 7B it recovers most of the retrieval gains without requiring hints. In WebShop, distillation matches or exceeds RAG while delivering higher scores with comparable or lower token cost.

Table 4: Performance of Qwen-2.5 14B Instruct and Qwen-2.5 7B Instruct models averaged across both ReAct and StateAct agents. See Appendix B.1 for individual agent performance.

| Model | Method | ALFWorld Success | WebShop Success | WebShop Score |
|-------|--------|------------------|-----------------|---------------|
| 14B | Base | 79.85% | 38.5% | 60.87 |
|  | Base+RAG | 82.09% | **43.5%** | 67.08 |
|  | SFT | 85.45% | 43.0% | 72.09 |
|  | Distilled (ours) | **91.04%** | **43.5%** | **72.40** |
| 7B | Base | 26.49% | 13.0% | 28.12 |
|  | Base+RAG | 71.27% | 8.5% | 18.46 |
|  | SFT | 62.69% | 22.0% | 54.38 |
|  | Distilled (ours) | **73.88%** | **22.5%** | **61.04** |

7B models benefit dramatically from hints (26.5% → 71.3%) in ALFWorld, but struggle to use them effectively at inference in WebShop. Hints push the small model towards the wrong attributes or over-constrain the search for the smaller models, the 14B model's higher capacity lets it better interpret hint semantics. Training offers a better solution, Distillation and SFT greatly improve performance compared to their incredibly low bases. The higher capacity lets the model interpret hint semantics and exploit them at test time better than the smaller models. Distillation compresses capability and, in WebShop-7B, matches a larger base (Distilled WebShop-7B 61.04 vs Base WebShop-14B 60.87). Across scales, the pattern is consistent: distillation recovers most of RAG's gains without permanent retrieval, with the largest efficiency advantage observed in WebShop.

## 5.2 EFFICIENCY ANALYSIS

Table 5 compares token usage and step counts, revealing that distilled models achieve superior performance with lower resource consumption. Distilled models use 10% fewer in ALFWorld and 47% fewer in WebShop vs. base, and 17–61% fewer vs. RAG, while taking fewer steps to complete tasks. This efficiency stems from cleaner action sequences learned from hint-improved teacher demonstrations.

Table 5: Efficiency and effectiveness of agents across environments using Qwen-2.5 14B Instruct. Metrics are averaged per episode. Success is used for ALFWorld, Score for WebShop. Lower is better for Tokens and Steps. See Appendix B.2 for 7B efficiency results.

| Environment | Method | Tokens/Episode ↓ | Steps/Episode ↓ | Success/Score ↑ |
|-------------|--------|------------------|-----------------|-----------------|
| ALFWorld | Base | 50.13k | 18.94 | 79.85% |
|  | RAG | 53.97k | 18.69 | 82.09% |
|  | SFT | 50.36k | 19.38 | 85.45% |
|  | Distilled (ours) | **44.82k** | **16.68** | **91.04%** |
| WebShop | Base | 7.99k | 7.16 | 60.87 |
|  | RAG | 11.05k | 6.34 | 67.08 |
|  | SFT | 4.29k | 5.00 | 72.09 |
|  | Distilled (ours) | **4.27k** | **4.98** | **72.40** |

As seen in Figure 3, across both environments, the trend is consistent. Retrieval helps, but its token overhead shifts methods to the right, while distillation recovers most of the retrieval gains at a much lower cost. In ALFWorld, distillation dominates the trade-off. It matches or exceeds RAG's accuracy at a fraction of the tokens, achieving the highest success rates (∼90-93%). In WebShop, the most efficient frontier is formed by both SFT and distillation, achieving the best scores (∼71–73) with the lowest average token counts per episode (below 6k). RAG improves over the base agent, but remains less token-efficient. The shaded ellipses highlight that these behaviors are regime-level,

not just single points: base runs generally cluster at low cost/low accuracy, RAG clusters at higher cost with high performance, and distillation clusters at low cost/highest performance, especially in ALFWorld.

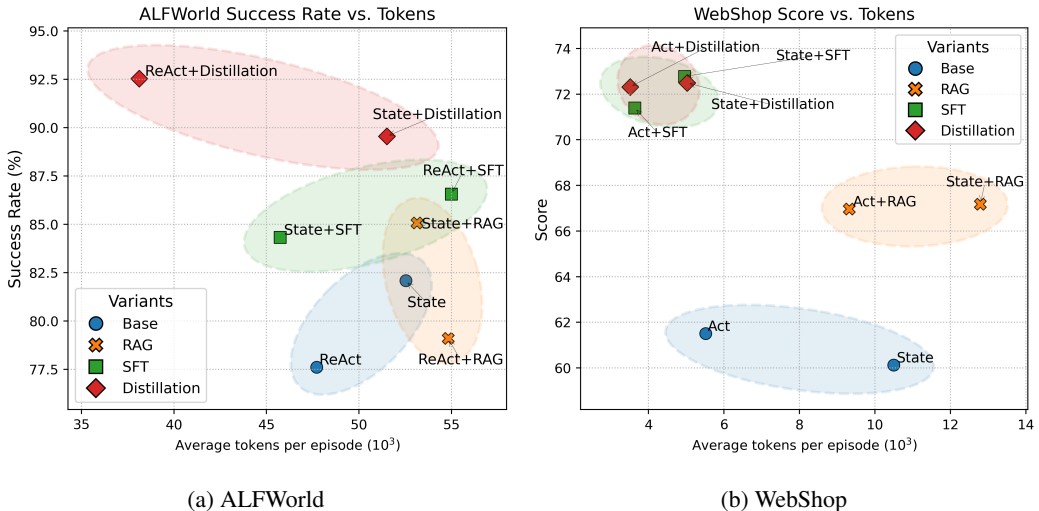

(a) ALFWorld

(b) WebShop

Figure 3: Accuracy–efficiency trade-off on ALFWorld and WebShop. The x-axis shows average tokens per episode, including retrieval overhead when used. Shapes denote training regime, Base (circle), RAG (cross), SFT (square), Distillation (diamond). Each point is a method variant (ReAct/StateAct). Shaded ellipses are provided to visually group variants. **Note:** Act refers to ReAct without "Thought" tokens.

## 6 LIMITATIONS

This study has several limitations. First, hint generation relies on repeated GPT-4o API calls, making cost dependent on the number of failures in the training data and potentially limiting scalability in larger environments. Second, retrieval is restricted to a single one-shot process at $t = 0$, which cannot adapt to mid-episode surprises and may be less effective in long-horizon or highly stochastic tasks. Third, all reported results are based on single-seed evaluation and therefore represent point estimates; while consistent trends are observed, multi-seed experiments are required to quantify variance and assess robustness. Finally, our evaluation is limited to the training environments (ALFWorld and WebShop), without testing cross-domain transfer. Understanding how well distilled agents generalize to novel settings remains an open question.

## 7 CONCLUSION

We presented a method for converting retrieval-augmented generation from a runtime necessity into a training-time teacher. By extracting hints from failures, using them to generate better trajectories, and distilling with hints removed, we produce agents that internalize guidance while eliminating deployment overhead. Distilled students achieve 91% success on ALFWorld (vs 79% baseline) and score 72.4 on WebShop (vs 60.9 baseline), while using fewer tokens than any alternative approach.

The method is simple, requires no expert supervision, and generalizes across model scales and agent architectures. Results suggest many augmentation strategies currently treated as runtime requirements might better serve as training-time supervision. Future work should explore dynamic retrieval triggers, trajectory-level objectives for long-horizon tasks, and cross-environment transfer to test whether distilled competencies truly generalize beyond their training distribution.

REPRODUCIBILITY STATEMENT

We have taken several steps to ensure reproducibility. A link to our anonymized source code, including training and evaluation scripts, is provided here: `https://anonymous.4open. science/r/anonymized-submission-iclr/README.md`. Detailed prompts are described in Appendix A. All experiments are run with fixed seeds. The processed hint banks and student training datasets used to generate the reported results are included in the source code. Together, these resources should enable full replication of the experiments and verification of our claims.

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

# APPENDIX

## A   PROMPTS

LLMs can be sensitive to prompt design, and the exact wording often plays a critical role in performance. To ensure full transparency and reproducibility, we include here the complete prompts used in our experiments. This appendix documents the prompts used for hint generation and agent evaluation.

### A.1   HINT GENERATION

The prompts used with GPT-4o to generate runtime hints are given below. For ALFWorld:

---

**Prompt for ALFWorld Hint Generation**

```
You are diagnosing why a household agent failed and creating
    runtime HINTS to avoid future failures in similar tasks.

Environment type: {env_type}
Task goal: {goal_txt}

=======
Steps before failure (action → observation):
{failure_trajectory}
=======

Emit STRICT JSON with this schema:
{{
  "hints": [
    {{
      "env_type": "{env_type}",
      "text": "≤120 chars, imperative advice the agent can follow in
            future for similar environment types"
    }}
  ]
}}

Rules:
- Focus on errors that explain THIS failure; provide hints to avoid
    failures on SIMILAR tasks.
- Make it generally applicable.
- Use placeholders like {{object}}, {{container}}, {{location}}, {{
    page}}, {{item}} instead of numbers/IDs.
- 1-4 high-value hints max. No duplicates. No meta commentary.
- JSON only. No extra text.
```

---

The prompt for WebShop is identical, it is given below:

```
Prompt for WebShop Hint Generation

You are diagnosing why a shopping agent failed and creating runtime
    HINTS to avoid future failures in purchasing similar items.

Item category: {category}
Task goal: {goal}

=======
Steps before failure (action → observation):
{failure_trajectory}
=======

Emit STRICT JSON with this schema:
{{
  "hints": [
    {{
      "category": "{category}",
      "text": "≤120 chars, imperative advice the agent can follow in
          future for similar environment types"
    }}
  ]
}}

Rules:
- Focus on errors that explain THIS failure; provide hints to avoid
    failures on SIMILAR tasks.
- Make it generally applicable.
- Use placeholders like {{item}}, {{size}}, {{color}}, {{attribute
    }}, {{material}} instead of names/IDs.
- 1-4 high-value hints max. No duplicates. No meta commentary.
- JSON only. No extra text.
```

## A.2 HINT RETRIEVAL PROMPT

Below are the prompts used in retrieval, they were designed to also account for usage mid-episode, but we ended up not using that functionality as it would increase token usage.

---

**Prompt for Hint Retrieval in ALFWorld**

```
You are selecting helpful hints for a household agent.
Choose up to {k} DISTINCT hints that are immediately useful for the
    current state.
If none apply, return an empty list.

Read the current goal, location, inventory and thought to avoid
    redundant/misleading hints.

===== Task & state =====
{query}

===== Hints List =====
1) {hint1}
2) {hint2}
...

Return STRICT JSON only (no prose):
'{"answer": [<indices from the list above>]}'

Do not include anything else.
```

---

**Prompt for Hint Retrieval in WebShop**

```
You are selecting helpful hints for a shopping agent.
Choose up to {k} DISTINCT hints that are immediately useful for the
    current state.
If none apply, return an empty list.

Read the current goal, location, inventory and thought to avoid
    redundant/misleading hints.

===== Task & state =====
{query}

===== Candidate hints (numbered) =====
1) {hint1}
2) {hint2}
...

Return STRICT JSON only (no prose):
'{"answer": [<indices from the list above>]}'

Do not include anything else.
```

A.3 AGENT PROMPT

Below are the prompts used for the main agent. It provides the agent with its task and few-shot examples, and includes a hint block (if using RAG). If evaluating without hints, the lines are removed from "Here are some hints:" until the separators "=============".

---

**Example Prompt for ALFWorld Agents (with RAG)**

```
Interact with a household to solve a task.

=============
Apply these rules silently to choose the next action.
Never repeat, quote, or paraphrase this block in thought or action.
If any rule conflicts with the current observation, prefer the
    observation.

Here are some hints:
- Identify and locate the {object} before attempting to move it.
- Ensure the {object} is cool before placing it in the {container}.
- Ensure the {object} is accessible and not obstructed by other
    items.

=============

Here are 2 examples:

You are in the middle of a room. Looking quickly around you, you
    see a cabinet 9, a cabinet 8, a cabinet 7, a cabinet 6, a
    cabinet 5, a cabinet 4, a cabinet 3, a cabinet 2, a cabinet 1, a
     coffeemachine 1, a countertop 3, a countertop 2, a countertop
    1, a drawer 9, a drawer 8, a drawer 7, a drawer 6, a drawer 5, a
     drawer 4, a drawer 3, a drawer 2, a drawer 1, a fridge 1, a
    garbagecan 1, a microwave 1, a shelf 3, a shelf 2, a shelf 1, a
    sinkbasin 1, a stoveburner 4, a stoveburner 3, a stoveburner 2,
    a stoveburner 1, and a toaster 1.
Your task is to: put a cool mug in shelf.
> think: To solve the task, I need to find and take a mug, then
    cool it with fridge, then put it in shelf.
OK.
> think: First I need to find a mug. A mug is more likely to appear
     in countertop (1-3), coffeemachine (1), cabinet (1-9), shelf
    (1-3), drawer (1-9). I can check one by one, starting with
    countertop 1.
OK.
> go to countertop 1
On the countertop 1, you see a knife 1, a pan 2, a pan 1, and a
    plate 1.
> go to countertop 2
On the countertop 2, you see a apple 2, a butterknife 2, a fork 3,
    a houseplant 1, a knife 2, a lettuce 3, a spatula 1, and a
    winebottle 1.
> go to countertop 3
On the countertop 3, you see a apple 1, a book 1, a bread 1, a
    butterknife 3, a glassbottle 2, a glassbottle 1, a knife 3, a
    lettuce 2, a lettuce 1, a papertowelroll 1, a plate 3, a spoon
    1, a statue 1, and a vase 2.
> go to coffeemachine 1
On the coffeemachine 1, you see nothing.
> go to cabinet 1
The cabinet 1 is closed.
> open cabinet 1
```

```
You open the cabinet 1. The cabinet 1 is open. In it, you see
    nothing.
> go to cabinet 2
The cabinet 2 is closed.
> open cabinet 2
You open the cabinet 2. The cabinet 2 is open. In it, you see
    nothing.
> go to cabinet 3
The cabinet 3 is closed.
> open cabinet 3
You open the cabinet 3. The cabinet 3 is open. In it, you see a pot
    1.
> go to cabinet 4
On the cabinet 4, you see nothing.
> go to cabinet 5
The cabinet 5 is closed.
> open cabinet 5
You open the cabinet 5. The cabinet 5 is open. In it, you see
    nothing.
> go to cabinet 6
The cabinet 6 is closed.
> open cabinet 6
You open the cabinet 6. The cabinet 6 is open. In it, you see a mug
    3.
> think: Now I find a mug (3). Next, I need to take it.
OK.
> take mug 3 from cabinet 6
You pick up the mug 3 from the cabinet 6.
> think: Now I take a mug (3). Next, I need to go to a fridge (1)
    and cool it.
OK.
> go to fridge 1
The fridge 1 is closed.
> cool mug 3 with fridge 1
You cool the mug 3 using the fridge 1.
> think: Now I cool the mug. Next, I need to put it in/on shelf 1.
OK.
> go to shelf 1
On the shelf 1, you see a cup 2, a cup 1, a peppershaker 1, and a
    saltshaker 1.
> put mug 3 in/on shelf 1
You put the mug 3 in/on the shelf 1.

You are in the middle of a room. Looking quickly around you, you
    see a cabinet 16, a cabinet 15, a cabinet 14, a cabinet 13, a
    cabinet 12, a cabinet 11, a cabinet 10, a cabinet 9, a cabinet
    8, a cabinet 7, a cabinet 6, a cabinet 5, a cabinet 4, a cabinet
     3, a cabinet 2, a cabinet 1, a coffeemachine 1, a countertop 2,
     a countertop 1, a diningtable 1, a drawer 5, a drawer 4, a
    drawer 3, a drawer 2, a drawer 1, a fridge 1, a garbagecan 1, a
    microwave 1, a safe 1, a sinkbasin 1, a stoveburner 4, a
    stoveburner 3, a stoveburner 2, a stoveburner 1, and a toaster
    1.
Your task is to: cool some pan and put it in stoveburner.
> think: To solve the task, I need to find and take a pan, then
    cool it with fridge, then put it in stoveburner.
OK.
> think: First I need to find a pan. An pan is more likely to
    appear in stoveburner (1-4), sinkbasin (1), diningtable (1),
    countertop (1-2), cabinet (1-16), drawer (1-5). I can check one
    by one, starting with stoveburner 1.
OK.
```

```
>go to stoveburner 1
On the stoveburner 1, you see nothing.
>go to stoveburner 2
On the stoveburner 2, you see a pot 1.
> go to stoveburner 3
On the stoveburner 3, you see a pan 1.
> think: Now I find a pan (1). Next, I need to take it.
OK.
> take pan 1 from stoveburner 3
You pick up the pan 1 from the stoveburner 3.
> think: Now I take a pan (1). Next, I need to go to a fridge (1)
    and cool it.
OK.
> go to fridge 1
The fridge 1 is closed.
> cool pan 1 with fridge 1
You cool the pan 1 using the fridge 1.
> think: Now I cool a pan (1). Next, I need to put it in/on
    stoveburner 1.
OK.
>go to stoveburner 1
On the stoveburner 1, you see nothing.
> put pan 1 in/on stoveburner 1
You put the pan 1 in/on the stoveburner 1.

============

Here is the task.
```

### Example Prompt for WebShop Agents (with RAG)

```
You are an intelligent WebShop assistant.
Your job is to interact with the environment using the `[]` buttons
    only.
You have 15 interactions to buy an item that is closest to the
    instruction.

============
Apply these rules silently to plan your actions.
Never repeat, quote, or paraphrase this block in thought or action.
If any rule conflicts with the current observation, prefer the
    observation.

Here are some hints:
- Ensure search terms include {size}, {color}, and {category} for
    precise results.
- Use specific attributes like {material} and {size} in search
    queries.
- Ensure search terms include {size}, {color}, and {price}
    constraints.
============

Here is 1 example:

Instruction:
i would like a 3 ounce bottle of bright citrus deodorant for
    sensitive skin, and price lower than 50.00 dollars
[Search]
</turn>
<turn>
Action: search[3 ounce bright citrus deodorant sensitive skin]
</turn>
<turn>
Observation:
[Back to Search]
Page 1 (Total results: 50)
[Next >]
[B078GWRC1J]
Bright Citrus Deodorant by Earth Mama | Natural and Safe for
    Sensitive Skin, Pregnancy and Breastfeeding, Contains Organic
    Calendula 3-Ounce
10.99
[B078GTKVXY]
Ginger Fresh Deodorant by Earth Mama | Natural and Safe for
    Sensitive Skin, Pregnancy and Breastfeeding, Contains Organic
    Calendula 3-Ounce
10.99
[B08KBVJ4XN]
Barrel and Oak - Aluminum-Free Deodorant, Deodorant for Men,
    Essential Oil-Based Scent, 24-Hour Odor Protection, Cedar &
    Patchouli Blend, Gentle on Sensitive Skin (Mountain Sage, 2.7 oz
    , 2-Pack)
15.95
</turn>
<turn>
Action: think[B078GWRC1J and B078GTKVXY are bright citrus deodorant
     less then 50 dollars. I can check B078GWRC1J first.]
</turn>
<turn>
Observation: OK.
</turn>
<turn>
```

```
Action: click[B078GWRC1J]
</turn>
<turn>
Observation:
[Back to Search]
[< Prev]
scent [assorted scents][bright citrus][calming lavender][ginger
    fresh][simply non-scents]
size [travel set (4-pack)][3 ounce (pack of 1)][3-ounce (2-pack)]
Bright Citrus Deodorant by Earth Mama | Natural and Safe for
    Sensitive Skin, Pregnancy and Breastfeeding, Contains Organic
    Calendula 3-Ounce
Price: 10.99
Rating: N.A.
[Description]
[Features]
[Reviews]
[Buy Now]
</turn>
<turn>
Action: think[For 3 ounce bottle of bright citrus deodorant for
    sensitive skin, the item has options 'bright citrus' and '3
    ounce (pack of 1)' and seems good to buy.]
</turn>
<turn>
Observation: OK.
</turn>
<turn>
Action: click[bright citrus]
</turn>
<turn>
Observation: You have clicked bright citrus.
</turn>
<turn>
Action: click[3 ounce (pack of 1)]
</turn>
<turn>
Observation: You have clicked 3 ounce (pack of 1).
</turn>
<turn>
Action: click[Buy Now]
</turn>

============

Here is the task.
```

# B EXTENDED RESULTS AND ABLATIONS

## B.1 PER-AGENT RESULTS

While the main text reports averages (Table 4), here we report per-agent results.

Tables 6 and 7 break down performance by agent family. For 14B models, both ReAct and State-Act benefit from distillation: ReAct shows the largest jump in ALFWorld (77.6% → 92.5%), while StateAct achieves the strongest WebShop performance (score 72.8–72.5). For 7B models, the same trend holds: ReAct+Distillation lifts WebShop score from 44.0 to 57.1, while StateAct+Distillation reaches 79.1% success on ALFWorld and 65.0 WebShop score, substantially outperforming its baselines. These per-agent breakdowns confirm that distillation is effective across prompting styles, with complementary strengths: ReAct gains most in ALFWorld, while StateAct consistently excels in WebShop.

Table 6: Performance of Qwen-2.5 14B Instruct agents across methods. ALFWorld reports Success (%), WebShop reports Success (%) and Score.

| Method (14B) | ALFWorld Success | WebShop Success | WebShop Score |
|---|---|---|---|
| ReAct | 77.61 | 37.00 | 61.61 |
| +RAG | 79.10 | **43.00** | 66.97 |
| +SFT | 86.57 | 42.00 | 71.40 |
| +Distillation (ours) | **92.54** | **43.00** | **72.32** |
| StateAct | 82.09 | 40.00 | 60.12 |
| +RAG | 85.07 | **44.00** | 67.18 |
| +SFT | 84.33 | **44.00** | **72.78** |
| +Distillation (ours) | **89.55** | **44.00** | 72.49 |

Table 7: Performance of Qwen-2.5 7B Instruct agents across methods. ALFWorld reports Success (%), WebShop reports Success (%) and Score.

| Method (7B) | ALFWorld Success | WebShop Success | WebShop Score |
|---|---|---|---|
| ReAct | 11.19 | 19.00 | 44.05 |
| +RAG | **73.13** | 9.00 | 28.80 |
| +SFT | 50.00 | 22.00 | 44.25 |
| +Distillation (ours) | 68.66 | **31.00** | **57.11** |
| StateAct | 41.79 | 5.00 | 12.18 |
| +RAG | 69.40 | 4.00 | 8.12 |
| +SFT | 75.37 | 38.00 | 64.51 |
| +Distillation (ours) | **79.10** | **39.00** | **64.97** |

## B.2 7B EFFICIENCY RESULTS

Table 8 compares token usage and step counts for 7B models. As with 14B (Table 5), distillation provides the strongest balance of effectiveness and efficiency. In ALFWorld, distilled students reach 68.7% success, far above the base (11.2%) and close to RAG (73.1%), while avoiding the heavy runtime dependence on retrieval. In WebShop, the advantage is even clearer: distillation improves the score to 57.1 while using fewer tokens than all other methods. Taken together, these results show that distillation allows 7B models to internalize most of the retrieval gains, achieving accuracy competitive with larger or retrieval-augmented baselines while being substantially more efficient.

Table 8: Efficiency and effectiveness of agents across environments using Qwen-2.5 7B Instruct. Metrics are averaged per episode. Success is used for ALFWorld, Score for WebShop. Lower is better for Tokens and Steps.

| Environment | Method | Tokens/Episode ↓ | Steps/Episode ↓ | Success/Score ↑ |
|---|---|---|---|---|
| ALFWorld | Base | **48.71k** | **22.26** | 11.19% |
| | RAG | 62.74k | 27.16 | **73.13%** |
| | SFT | 62.08k | 25.14 | 50.00% |
| | Distilled (ours) | 67.62k | 27.86 | 68.66% |
| WebShop | Base | 9.36k | 7.89 | 44.05 |
| | RAG | 22.02k | 10.07 | 28.80 |
| | SFT | 8.18k | 8.50 | 44.25 |
| | Distilled (ours) | **5.95k** | **7.15** | **57.11** |

## B.3  RETRIEVAL DEPTH (K) ABLATIONS

In our experiments, we initially set k=3 as a pragmatic default. Small k is preferred in retrieval-augmented setups because it balances performance against prompt length. To validate this choice, we swept $k \in \{1, 3, 6, 9\}$ on both ALFWorld and WebShop with Qwen-14B and report success, score, steps, and token cost (Tables 9 & 10).

Table 9: Different retrieval k on ALFWorld with Qwen-2.5 14B Instruct. Results are averaged across ReAct and StateAct agents.

| Top-k | Success Rate (%) | Avg. Steps | Avg. Tokens |
|---|---|---|---|
| 1 | 83.96 | 19.11 | 52.02k |
| 3 (ours) | 82.09 | 18.69 | 53.97k |
| 6 | **84.33** | **18.13** | **50.95k** |
| 9 | 76.87 | 19.27 | 57.26k |

For ALFWorld, performance is stable across $k \in \{1, 3, 6\}$, with success rates in the 82–84% range and token cost between 51k–54k. A sharp drop appears at k=9, where success falls to 76.9% and token usage inflates to over 57k tokens. This confirms that large hint blocks become counterproductive by increasing prompt length without improving guidance.

In WebShop, retrieval depth also shows diminishing returns. The best balance is reached at k=3 (43.5% success, score 67.1), which ties or slightly improves on k=1 while maintaining similar token cost. Larger k values (6 and 9) lead to higher token usage (12–13k) and reduced scores ($\approx$61), showing that excessive hints can over-constrain search.

Table 10: Different retrieval k on WebShop with Qwen-2.5 14B Instruct. Results are averaged across ReAct and StateAct agents.

| Top-k | Success Rate (%) | Score | Avg. Steps | Avg. Tokens |
|---|---|---|---|---|
| 1 | 40.50 | 64.73 | 6.45 | **11.04k** |
| 3 (ours) | **43.50** | **67.08** | **6.34** | 11.05k |
| 6 | 40.00 | 61.80 | 6.94 | 12.75k |
| 9 | 40.50 | 61.12 | 7.16 | 13.12k |

We therefore keep k=3 as a uniform default. This beats k=1 in WebShop, remains competitive with k=1 and 6 in ALFWorld, and avoids the instability of k=9 in both environments. If we were to tune

per domain, a simple policy would be to use more hints (k=6) for structure-heavy tasks (ALFWorld), and use fewer hints (k=3) for noisy, attribute-heavy tasks (WebShop).

## C  THE USE OF LARGE LANGUAGE MODELS (LLMs)

We acknowledge using ChatGPT (GPT-5) provided by OpenAI (`https://chatgpt.com`) to help draft, edit text, and to produce LaTeX code templates. All outputs were reviewed, fact-checked, and substantially edited by us. We confirm that the work presented is our own.

