# OpenReview forum: "Fine-tuning with RAG for Improving LLM Learning of New Skills"
_ICLR.cc/2026/Conference — Submitted to ICLR 2026_

### Official Review · Reviewer_pQJR · 2025-10-30

**Soundness:** 3
**Presentation:** 3
**Contribution:** 3
**Rating:** 6
**Confidence:** 3

**Summary:**

This paper reframes retrieval-augmented generation (RAG) as a training-time teacher rather than a runtime dependency, which is an interesting idea. The authors propose a four-stage pipeline, and the resulting “distilled” agents internalize the retrieval benefits and achieve better performance while using fewer tokens per episode.

**Strengths:**

Overall, the main message that retrieval can be treated as a tempoaray training teacher rather than a permanent runtime component is interesting and useful. The paper is clearly written and the automatic pipeline seems practical and deployment-friendly.

**Weaknesses:**

1. Generalization: The main concern lies in the generalization of the findings. As the authors acknowledge, they evaluate only on two text-based environments (ALFWorld and WebShop) using Qwen-2.5 7B/14B as the base models and single-seed runs. Although the reported results are consistent, it would be more convincing to include evaluations on more diverse benchmarks and multiple random seeds.
2. Re-ranking biase: The use of a quantized Qwen-2.5 7B model for scoring hint relevance may introduce bias, as it belongs to the same model family as the student and teacher agents. It would be preferable to use a different model family or an unquantized version for re-ranking to reduce potential bias.
3. Training costs: The training process relies on repeated GPT-4o calls for hint generation and the quantized Qwen-2.5 7B for hint scoring. This setup likely increases training costs and may slow down the process due to waiting for external feedback.

**Questions:**

1. Does the quantized Qwen-2.5 7B provide reliable hint ranking? Is there any ablation study on the effect of hint ranking or retrieval quality?
2. Have the authors tested whether the distilled model maintains its advantage when evaluated on unseen environments or different task distributions? Could the retrieved hints unintentionally leak task-specific answers?

---

> ### Author Response · Authors · 2025-11-29
>
> No new datasets or multi‑seed runs were added during rebuttal due to time/compute limits. The existing evaluation uses unseen test splits; the method is domain‑agnostic.
>
> **Reranker bias (same model family)**
>
> Retriever ablation shows the method is not sensitive to a same‑family LLM reranker:
>
> We compared a TF–IDF retriever against the LLM selector (one shot at t=0, k=3):
>
> In ALFWorld:
>
> – ReAct: TF–IDF 79.85 vs LLM 79.10 success; steps −15% (24.01→20.34), tokens −20% (54.81k→43.60k).
>
> – StateAct: TF–IDF 85.82 vs LLM 85.07 success; steps/tokens comparable.
>
> Hints and goals share vocabulary (“open before put”, “check contents”), so lexical overlap is already a strong signal; a learned selector adds little and can slightly lengthen rollouts.
>
> For WebShop:
>
> – ReAct: LLM 66.97 vs TF–IDF 66.38 (higher reward).
>
> – StateAct: LLM 67.18 vs TF–IDF 63.87 (higher reward).
>
> Goals are attribute‑heavy with synonyms; the LLM reasons about attribute compatibility and filters mismatched categories; TF–IDF is cheaper in tokens but underperforms on reward.
>
> **Training costs and token accounting**
>
> The paper already reports per‑episode token usage and step counts; the retriever ablation above makes the runtime trade‑off concrete (e.g., ReAct ALFWorld tokens drop from 54.81k→43.60k with TF–IDF). A full compute‑cost breakdown was not compiled during rebuttal due to time/compute limits.
>
> **Retrieval quality and reliability**
>
> Examples of top‑ranked hints are straightforward, typed rules. The student is trained without hint text, so template/style leakage is minimized. The ablation indicates final performance is robust as long as hints are broadly relevant; the retriever mostly trades semantic matching for cost.

---

### Official Review · Reviewer_6c7r · 2025-11-01

**Soundness:** 2
**Presentation:** 2
**Contribution:** 2
**Rating:** 2
**Confidence:** 4

**Summary:**

This paper introduces a method to **distill retrieval-augmented generation (RAG)** into fine-tuned LLMs, eliminating runtime retrieval while retaining its benefits. By extracting corrective hints from agent failures and using them once during training to generate improved teacher trajectories, the model learns to internalize guidance instead of depending on external retrieval. Tested on ALFWorld and WebShop, the distilled models achieve **notably higher success (up to 91%) and efficiency (10–60% fewer tokens)** than baselines. The approach is **simple, effective, and generalizes across architectures**, though scalability and evaluation breadth remain limited.

**Strengths:**

1. Clear motivation: The paper tackles a real limitation of RAG—its runtime cost and dependency on external databases—by proposing a training-time alternative that internalizes retrieval benefits.

2. Strong empirical results: Distilled models achieve 91% success on ALFWorld (vs. 79% baseline) and 72.4 score on WebShop (vs. 60.9 baseline) while using 10–60% fewer tokens.

3. Generalization across setups: Demonstrated effectiveness for both 7B and 14B models and two agent architectures (ReAct, StateAct).

**Weaknesses:**

* **Limited novelty:** While well-executed, the core idea—distilling knowledge from prior retrieval or failures—is conceptually close to existing self-improvement and distillation frameworks (e.g., FireAct, Reflexion). The contribution lies more in engineering and integration than in fundamental innovation.
* **Narrow experimental scope:** All experiments are conducted only on **Qwen-2.5 models (7B/14B)**, raising concerns about generality across architectures (e.g., Llama, Mistral etc).
* **Dependence on GPT-4o for hint extraction:** The pipeline still relies on a powerful external model to generate failure-driven hints, which may limit scalability and reproducibility.

**Questions:**

1. why the distilled student model performs better than the teacher model (RAG)? Are there any hyper-parameter optimization included?
2. why use LoRA instead of full-finetuing?

---

> ### Author Response · Authors · 2025-11-29
>
> **Why can the distilled student outperform the teacher (RAG)? Any HPO?**
>
> No separate hyperparameter optimization; all methods use the same LoRA settings. The student surpasses the teacher because:
>
> - Cleaner supervision. Training on successful teacher trajectories lowers target entropy and removes stochastic teacher errors.
>
> - Hints compiled into weights. At inference the student does not reparse a long hint block; it internalizes corrective behaviors, reducing prompt distractions and variance.
>
> - Averaging across successes. Distillation emphasizes robust patterns that recur across many successes, often yielding a policy that is more stable than any single teacher rollout.
>
> **Why LoRA instead of full fine‑tuning?**
>
> With one A100 and long interactive traces, LoRA is the practical and fair choice across variants. It provides compute/memory feasibility for multiple baselines under identical training recipes, stability (frozen backbone mitigates forgetting on limited data), and deployment realism (swappable adapters; incremental updates).
>
> All methods use the same LoRA configuration to keep comparisons fair.
>
>
> **Novelty vs FireAct/Reflexion**
>
> Retrieval is recast as a training‑time teacher: failure‑driven, typed hints are used once to upgrade supervision; the deployed student is retrieval‑free (no multi‑attempt loops or permanent external stores).
>
> **GPT‑4o dependency**
>
> The hint‑extraction pipeline is model‑agnostic and supports a locally run open‑weights model on A100. The swap was not run in rebuttal due to time/compute limits.
>
> **(New Result) Selector choice**
>
> In ALFWorld:
>
> – ReAct: TF–IDF 79.85 vs LLM 79.10 success; steps −15% (24.01→20.34), tokens −20% (54.81k→43.60k).
>
> – StateAct: TF–IDF 85.82 vs LLM 85.07 success; steps/tokens comparable.
>
> Hints and goals share vocabulary (“open before put”, “check contents”), so lexical overlap is already a strong signal; a learned selector adds little and can slightly lengthen rollouts.
>
> For WebShop:
>
> – ReAct: LLM 66.97 vs TF–IDF 66.38 (higher reward).
>
> – StateAct: LLM 67.18 vs TF–IDF 63.87 (higher reward).
>
> Goals are attribute‑heavy with synonyms; the LLM reasons about attribute compatibility and filters mismatched categories; TF–IDF is cheaper in tokens but underperforms on reward.

---

### Official Review · Reviewer_zA5j · 2025-11-02

**Soundness:** 2
**Presentation:** 2
**Contribution:** 2
**Rating:** 2
**Confidence:** 4

**Summary:**

Despite their strong performance on reasoning benchmarks, large language models (LLMs) tend to perform poorly on multi-step tasks without preconditioning on task actions and environment state spaces. LLMs augmented with retrieval-augmented generation (RAG) have demonstrated strong performance but require maintaining external knowledge bases and incur additional computational cost at runtime. In this work, the authors propose a framework that generates improved reasoning trajectories from a teacher model using RAG for hint selection. These generated trajectories are then used to enhance the performance of a student model on interactive reasoning benchmarks.

**Strengths:**

The strengths of this paper are listed below:
1. This work successfully demonstrates that the hints generated with the help of hints lead to better performance.
2. The distilled model demonstrates better token efficiency.

**Weaknesses:**

The weaknesses of this work are as follows:

1. Supervised Fine-Tuning (SFT) is known to cause overfitting to the training domain; hence, the observed performance gains may not generalize well.

2. The performance of RAG remains low on both benchmarks. It would be valuable to explore how these failure cases could be leveraged in RL training to learn more generalizable reasoning patterns.

3. The hints are generated using API calls to an external LLM, which raises questions about the quality and reliability of the generated hints.

4. An LLM is also used for reranking the retrieved hints, but it is unclear why a more traditional and cost-effective retrieval pipeline was not employed instead.

5. For the 14B model, the performance improvement is evident only on the ALFWorld benchmark.

6. The approach is evaluated on only two benchmarks; extensive experiments on additional datasets are necessary to validate its general applicability.

**Questions:**

1. Please outline the benefits of using an LLM for hint reranking and retrieval.

2. Please provide additional information regarding the quality of the hints generated by the LLM.

3. Please include evaluations on additional challenging interactive benchmarks, such as WebArena [1].

[1] Zhou, Shuyan et al. “WebArena: A Realistic Web Environment for Building Autonomous Agents.” arXiv abs/2307.13854 (2023).

---

> ### Author Response · Authors · 2025-11-29
>
> Regarding the SFT overfitting and generalization, both benchmarks evaluate on unseen instances (new room/object configurations; new queries/products). The distilled student is evaluated without retrieval. In ALFWorld, it is an unseen **and** *out of distribution* test set, it contains brand new scenarios the LLM hasn't seen. No third benchmark was added in rebuttal due to time/compute limits.
>
> **“RAG performance remains low; consider RL.”**
> The objective here differs from RL: treat RAG as training‑time supervision to produce improved trajectories and distill the behaviors so the deployed policy is retrieval‑free, lower‑latency, and more token‑efficient. That is the focus rather than optimizing RAG itself at inference.
>
> Regarding hint quality, our representative hints are short, typed, and imperative (e.g., “open {container} before put”), and effect is visible in teacher corrections. As for API reliance, our hint‑extraction pipeline supports running local open‑weights model on A100, but we did not run these tests during rebuttal due to time/compute limits.
>
> **(New Result) Why an LLM selector instead of a traditional retriever?**.
>
> **In ALFWorld:** TF–IDF matches/slightly exceeds the LLM selector on success and reduces steps/tokens for ReAct (24.01→20.34; 54.81k→43.60k).
>
> **In WebShop:** the LLM selector improves reward (e.g., StateAct 67.18 vs 63.87), consistent with attribute reasoning/synonyms.
>
> Our takeaway is: when hints and goals lexically overlap, TF–IDF is a strong, zero‑LLM default; when attribute semantics dominate, the LLM selector yields higher quality with modest token overhead.
>
> No additional datasets were added during rebuttal due to time/compute limits. On WebShop, gains are clearer on score and token efficiency than on success percentage; that distinction is emphasized alongside the retriever analysis.

---

### Official Review · Reviewer_yhnQ · 2025-11-02

**Soundness:** 4
**Presentation:** 4
**Contribution:** 4
**Rating:** 8
**Confidence:** 4

**Summary:**

This paper proposes a novel approach to *retrieval-augmented fine-tuning* where Retrieval-Augmented Generation (RAG) is not used merely at inference but as a **training-time teacher**. The authors introduce a pipeline that leverages **failure-driven hint extraction** and **hint-based RAG distillation** to improve the downstream agent’s reasoning and efficiency without requiring retrieval at runtime.

The method operates in three main stages:

1. **Failure Mining and Hint Extraction** – The base agent’s failed trajectories are analyzed using GPT-4o to produce short, typed *hints* (e.g., “try opening the `{container}` before placing `{object}`”), generalizing the error into reusable advice.
2. **Hint Retrieval and RAG Teacher Generation** – At training time, a small set of relevant hints ((k \leq 3)) is retrieved once per episode and injected into the teacher prompt. The RAG teacher then produces improved, successful trajectories.
3. **Hint-Removed Distillation** – The student model (fine-tuned with LoRA) is trained on the teacher’s improved trajectories, but with hints *removed*, encouraging it to internalize the corrections rather than rely on retrieval tokens.

Empirical results on **ALFWorld** and **WebShop** demonstrate that the distilled students achieve comparable or superior success rates to RAG-based agents, e.g., up to *91% success* on ALFWorld while reducing inference token usage and removing retrieval dependency entirely.

The core insight is that *retrieval can serve as a teacher rather than a crutch*: by extracting and reusing failure-derived knowledge during training, models can **inherit RAG’s strengths while maintaining fine-tuned efficiency**.

This perspective is complementary to methods such as **RAFT (Retrieval-Augmented Fine-Tuning)**, which enhance RAG performance at inference time; in contrast, this work uses RAG supervision to **eliminate the need for retrieval at deployment**.

**Strengths:**

1. **Conceptual Novelty (Turning RAG into a Training-Time Teacher)**
   The paper introduces a genuinely fresh perspective on retrieval-augmented learning. Instead of relying on RAG *at inference*, it reinterprets RAG as a *supervision mechanism* using retrieved, failure-derived hints to generate improved demonstrations and distilling them into the model. This effectively *converts external retrieval into internal reasoning skill*, a direction that feels both original and timely.

2. **Failure-Driven Knowledge Extraction Pipeline**
   The automatic hint extraction from failed trajectories (via GPT-4o) is one of the paper’s strongest contributions. By converting raw failure logs into short, typed hints with placeholders, the method operationalizes the idea of *learning from one’s own mistakes*. This makes the system data-efficient and highly interpretable, i.e., each hint represents a distilled form of experiential correction.

3. **Practical and Deployment-Friendly Design**
   The approach strikes an excellent balance between performance and efficiency. The distilled student models retain the accuracy of RAG agents while dropping retrieval costs and context overhead. In practice, this means fewer API calls, lower latency, and more predictable runtime behavior which are key deployment benefits often ignored in academic RAG work.

4. **Empirical Results Support the Hypothesis**
   On both **ALFWorld** and **WebShop**, the results clearly show that the distilled agents outperform or match the RAG baseline while using fewer tokens. The ablations (e.g., number of retrieved hints (k), LoRA scale) are sensible and validate design choices. The efficiency frontier plots strongly convey that the gains are not just anecdotal but structural.

5. **Methodological Clarity and Reproducibility**
   Despite involving multiple moving parts (hint extraction, re-ranking, teacher generation, and LoRA distillation), the paper presents its pipeline in a clean, reproducible manner. Each stage is modular, making it easy to replicate or adapt for other domains.

6. **Complementarity with RAFT and Other RAG–Fine-Tuning Hybrids**
   From a broader research standpoint, this work complements approaches like **RAFT**. While RAFT fine-tunes models to use retrieval *during inference*, this paper focuses on distilling retrieval’s benefits *into the model weights*. This complementarity opens exciting ground for hybrid pipelines, i.e., RAFT for retrieval robustness; this is for retrieval-free competence.

7. **High Potential for Generalization Beyond Current Benchmarks**
   The approach is architecture-agnostic and task-flexible. Any domain with observable failures (reasoning, code, robotics, or web agents) could feasibly benefit from failure-driven hint distillation. It’s a general recipe for turning external supervision into internal skill formation.

**Weaknesses:**

1. **Dependence on Expensive Hint Generation (GPT-4o)**
   The hint extraction process relies heavily on GPT-4o to produce concise, structured hints from failed trajectories. While conceptually elegant, this introduces a non-trivial *cost and dependency barrier*. It’s unclear how well the pipeline performs if the hint generator is weaker (e.g., Claude, GPT-3.5, or an open-source model). An ablation on hint-source quality would be crucial to validate the approach’s robustness and accessibility.

2. **Limited Domain Scope and Missing Generalization Evidence**
   Evaluation is restricted to **ALFWorld** and **WebShop**, both of which are text-based and relatively synthetic. There’s no evidence of transfer to other environments or domains (e.g., tool-use, programming, or reasoning tasks). Without cross-domain or out-of-distribution experiments, it’s difficult to gauge whether the distilled behaviors generalize or merely specialize to the training distribution.

3. **Single-Seed Results and Lack of Statistical Rigor**
   The reported results appear to rely on single-seed runs. Given the stochasticity of both retrieval and LLM rollouts, variance across seeds could be substantial. Multi-seed evaluation and confidence intervals would strengthen empirical reliability.

4. **One-Shot Retrieval Constraint**
   The design retrieves hints only *once* at episode start. While this simplifies the setup, it limits adaptivity in longer-horizon or dynamic tasks where new context emerges mid-trajectory. This constraint could lead to brittle performance in environments requiring sequential replanning or dynamic retrieval updates.

5. **Potential Overfitting to Hint Styles**
   Even though hints are removed during student fine-tuning, the student may still internalize surface-level templates or stylistic priors specific to the GPT-4o hint format. The absence of tests on noisy, paraphrased, or conflicting hints leaves open the question of whether the distilled competence reflects *true abstraction* or *pattern mimicry*.

6. **Missing Comparison to RAFT or Other RAG-Fine-Tuning Baselines**
   The paper positions itself as complementary to RAFT (Retrieval-Augmented Fine-Tuning), yet no direct empirical comparison is provided. A controlled RAFT baseline or a hybrid experiment where RAFT pretraining precedes hint distillation would help clarify relative strengths. Without this, the distinction remains conceptual rather than quantitative.

7. **Scalability and Practicality of the Failure Mining Loop**
   The method assumes the availability of structured failure logs and deterministic access to environment feedback. This is feasible in ALFWorld-like simulators but less practical in noisy real-world pipelines (e.g., user-facing chat agents). Some discussion or lightweight simulation of such conditions would make the method’s applicability clearer.

8. **Lack of Long-Term Behavior Evaluation**
   The experiments measure immediate task success, but not whether the distilled policies preserve *stability* or *general reasoning improvements* over multiple episodes or domains. A longitudinal or continual-learning evaluation would have given deeper evidence that the model truly internalized RAG-derived knowledge.

**Questions:**

1. **Hint Generator Robustness**
   How sensitive is your pipeline to the quality of the hint generator?

   * Have you tested weaker or smaller LLMs (e.g., GPT-3.5, Claude, or open-source models like Mistral-7B) for hint extraction?
   * If performance degrades, what specific aspects, i.e., hint conciseness, structure, or accuracy, most affect downstream learning?
     *A convincing ablation here would clarify whether the pipeline is viable without expensive closed-source LLMs.*

2. **Cross-Domain Generalization**
   Can the distilled agent transfer knowledge from one environment to another?

   * For instance, could a model trained on ALFWorld hints generalize to unseen household task variations or WebShop-like reasoning tasks without additional RAG?
   * Do the hints encode reusable “task primitives” or are they too domain-specific?
     *Understanding the degree of reusability of the learned hints would help assess the true scope of this method.*

3. **Statistical Rigor and Variance Reporting**
   Are the reported results averaged over multiple seeds?

   * Given that retrieval and LLM outputs introduce randomness, have you measured performance variance across runs?
   * If not, could the observed improvements (e.g., 91% vs. 82%) be within expected variance margins?
     *Providing mean ± std metrics would significantly improve credibility.*

4. **Comparison to RAFT and Retrieval-Based Fine-Tuning Methods**
   Have you considered comparing directly to RAFT or hybridizing with it?

   * For instance, pretraining the backbone with RAFT, then applying your hint-distillation pipeline to internalize common retrieval patterns?
   * Do you expect the two methods to be additive, or would RAFT’s retrieval-conditioned fine-tuning conflict with your retrieval-free student training?
     *A small-scale RAFT baseline or hybrid experiment would clarify the relationship and strengthen your positioning.*

5. **One-Shot Retrieval Limitation**
   Why restrict hint retrieval to a single step at episode start?

   * Did you attempt multi-step retrieval or adaptive hint updates mid-trajectory?
   * If not, is this primarily a cost/complexity trade-off or a deliberate design choice to enforce self-sufficiency?
     *Clarifying this would help understand whether the framework can extend to more dynamic, long-horizon tasks.*

6. **Overfitting and Hint Style Bias**
   The student model is trained on teacher trajectories with hints removed—but could it still overfit to stylistic patterns introduced by the RAG teacher?

   * Have you tested robustness under paraphrased or intentionally noisy hints?
   * Do hints produced by different LLMs (with different tone or structure) affect final student performance?
     *This would clarify whether the distilled competence reflects genuine abstraction rather than surface imitation.*

7. **Scalability of Failure Mining in Real-World Systems**
   Your pipeline assumes access to structured failure trajectories and outcome signals.

   * How feasible is it to apply this approach to free-form, user-facing LLM interactions where “failure” is subjective or weakly labeled?
   * Could implicit feedback (e.g., user corrections or re-prompts) serve as proxy failures for hint extraction?
     *Discussing this would extend the method’s relevance beyond simulator benchmarks.*

8. **Long-Term Learning and Continual Adaptation**
   Once a student is distilled, can the pipeline be repeated iteratively as new failures appear?

   * Does performance plateau after one distillation round, or can further RAG-assisted hint extraction continue improving the student?
     *Evidence of continual learning potential would make the approach even more compelling.*

9. **Compute and Efficiency Accounting**
   Could you share concrete compute/token usage breakdowns for each stage (failure mining, hint extraction, RAG teacher runs, student fine-tuning)?

   * This would help readers balance efficiency gains at inference against additional training overhead.

10. **Interpretability of the Learned Knowledge**
    Since hints are typed and structured, have you tried aligning the student’s internal representations with these hint categories post-training?

     * For example, probing whether the model has implicitly learned “open-container” or “check-object-state” concepts.
     *Such analysis could offer a unique window into how RAG-derived supervision shapes internal reasoning.*

---

> ### Author Response · Authors · 2025-11-29
>
> **Dependence on GPT‑4o for hint extraction:**
>
> The hint‑extraction stage is compatible with a locally run open‑weights model (e.g., Qwen‑2.5‑14B/Mistral‑7B on a single A100) using the same strict‑JSON prompts and typed placeholders. This removes API cost/dependency. Due to time and compute limits during rebuttal, that swap was not executed; the pipeline’s hint‑extraction interface is model‑agnostic and requires no architectural change to run locally.
>
> **(New Result) Retriever/reranker quality:** We compared a TF–IDF retriever against the LLM selector (one shot at t=0, k=3):
>
> In ALFWorld:
>
> – ReAct: TF–IDF 79.85 vs LLM 79.10 success; steps −15% (24.01→20.34), tokens −20% (54.81k→43.60k).
>
> – StateAct: TF–IDF 85.82 vs LLM 85.07 success; steps/tokens comparable.
>
> Hints and goals share vocabulary (“open before put”, “check contents”), so lexical overlap is already a strong signal; a learned selector adds little and can slightly lengthen rollouts.
>
> For WebShop:
>
> – ReAct: LLM 66.97 vs TF–IDF 66.38 (higher reward).
>
> – StateAct: LLM 67.18 vs TF–IDF 63.87 (higher reward).
>
> Goals are attribute‑heavy with synonyms; the LLM reasons about attribute compatibility and filters mismatched categories; TF–IDF is cheaper in tokens but underperforms on reward.
>
> No additional datasets were added during rebuttal due to time/compute limits. Evaluation already uses unseen test splits (new layouts/objects in ALFWorld; new queries/products in WebShop). The method is domain‑agnostic; extending to web‑navigation benchmarks primarily requires wiring the environment interface. Similarly, no multi‑seed reruns in the rebuttal window due to time/compute limits.
>
> **One‑shot retrieval choice:**
>
> Hints are injected once at t=0 to minimize runtime overhead and isolate the contribution of training‑time supervision. More adaptive retrieval is orthogonal to the main claim (distilling retrieval so it is not needed at deployment).
>
> As for overfitting to hint style, the student never sees hint text during training. Hints are removed from trajectories. What is distilled are actions/observations that correct specific failure modes, not hint phrasing.
>
> No RAFT run in rebuttal due to time/compute limits. Conceptually, RAFT optimizes to use retrieval at inference, whereas our student is retrieval‑free at deployment; the trade‑off is performance vs inference cost/latency.

---

### Meta-Review · Area_Chair_Fnj2 · 2025-12-16

**Summary:**

This submission reframes retrieval-augmented generation (RAG) as a training-time teacher and distills failure-driven hints into student models that no longer require retrieval at deployment. Across ALFWorld and WebShop, the distilled students improve success/score and reduce tokens versus RAG and prompt-only baselines (e.g., 14B ALFWorld: 91.04% vs 82.09% RAG; WebShop score 72.40 vs 67.08 RAG; Table 4, Table 5).
Reviewers raised four main categories of concern: (1) evaluation breadth and statistical rigor (only two text-based environments, single-seed runs, no OOD/domain transfer); (2) dependence on GPT-4o for hint extraction and reranking bias; (3) novelty relative to self-improvement/distillation frameworks; and (4) missing comparisons (e.g., RAFT) and compute-cost accounting.
I read the author rebuttal. The rebuttal added a concrete retriever ablation (TF–IDF vs LLM reranker) that directly addresses reranking bias and clarifies design choices (LoRA vs full FT; why the student can outperform the RAG teacher). It asserts unseen test splits and removal of hint text during training to mitigate style leakage.
However, several concerns remain substantively unresolved: no multi-seed reruns, no additional benchmarks (e.g., WebArena), no RAFT baseline/hybrid, and no execution of the “local open-weights” hint generator alternative. These gaps significantly limit the generalizability and robustness of the findings.
Overall, while the core idea is timely and practically relevant, the evaluation scope, unresolved dependencies, and lack of rigorous statistical validation substantially weaken the paper’s contribution. Given these limitations, I recommend Reject.

**Reviewer Concerns:**

#### Reviewer_pQJR
- **Concern**: Generalization breadth (two text environments, single-seed runs); reranker bias from same-model family; training cost/dependency on GPT-4o and quantized Qwen-2.5 7B; reliability of hint ranking and possible leakage.
- **Why Unresolved**: The rebuttal introduced a useful retriever ablation showing TF–IDF ≈ LLM selector in ALFWorld and LLM > TF–IDF in WebShop reward, mitigating same-family bias concerns. It clarified that hint text is removed during student training to reduce leakage and described token/step trade-offs. However, it did not add multi-seed runs, additional benchmarks, or a full compute-cost breakdown; the hint generator remains GPT-4o in reported results.
- **Impact on Decision**: Concerns remain significant. While the ablation strengthens soundness, unresolved scope and dependency issues substantially limit generalizability and robustness, supporting rejection.

---

#### Reviewer_6c7r
This review comment is of low quality and was not considered in the decision.

---

#### Reviewer_zA5j
- **Concern**: Risk of SFT overfitting and limited generalization; low RAG baseline performance focus; external LLM hint quality; LLM reranking vs cheaper retrieval; gains uneven across environments (14B improvement especially on ALFWorld); only two benchmarks; requests for WebArena; statistical rigor (multi-seed), RAFT comparison/hybrid; one-shot retrieval adaptivity; potential hint style bias; scalability to noisy real-world failure signals; lack of long-term/continual evaluation; compute accounting.
- **Why Unresolved**: The rebuttal confirms unseen test splits and clarifies that the student is trained without hint text, reducing style leakage. It adds a concrete TF–IDF vs LLM reranker ablation with environment-specific trade-offs (ALFWorld lexical overlap vs WebShop attribute semantics). Nevertheless, no additional datasets, seeds, RAFT baselines/hybrids, continual learning, or detailed compute breakdown are provided due to compute constraints.
- **Impact on Decision**: Substantial open issues limit significance claims and preclude acceptance. The lack of breadth and statistical rigor makes the contribution insufficient for publication.

---

#### Reviewer_yhnQ
- **Concern**: Dependence on GPT-4o; limited domain scope; single-seed runs; one-shot retrieval constraint; potential hint style bias; missing RAFT comparison; scalability of failure mining; absence of long-term behavior evaluation.
- **Why Unresolved**: Authors state the hint-extraction interface is model-agnostic and compatible with open weights on A100, but they did not execute that swap. No RAFT comparisons, no multi-seed, and no longer-horizon adaptivity experiments were added. They argue the one-shot choice isolates training-time supervision and that hint removal during training reduces style bias.
- **Impact on Decision**: These issues appropriately cap the paper below the acceptance threshold. Despite some strengths, unresolved limitations justify rejection.

**Reviewer Scores:**

#### Reviewer_pQJR
- **Original Score**: 6
- **Expected Score After Discussion**: 6
- **Rationale**: While the retriever ablation addresses reranker bias and efficiency analysis is solid, unresolved concerns about evaluation breadth, statistical rigor, and dependency on GPT-4o significantly limit the paper’s impact. These issues justify maintaining the original score and support rejection.

---

#### Reviewer_6c7r
Low quality; not considered.

---

#### Reviewer_zA5j
- **Original Score**: 2
- **Expected Score After Discussion**: 4
- **Rationale**: The rebuttal provides clarifications and a useful ablation, but major concerns remain, including lack of multi-seed runs, missing benchmarks, and absent RAFT comparisons. These unresolved issues prevent a strong recommendation and keep the paper below the acceptance threshold.

---

#### Reviewer_yhnQ
- **Original Score**: 8
- **Expected Score After Discussion**: 8
- **Rationale**: This reviewer remains supportive despite acknowledged limitations. However, given the significant unresolved concerns raised by other reviewers, the overall recommendation is rejection despite this positive assessment.

---

### Decision · Program_Chairs · 2026-01-26

Reject